# Inadequate BiP availability defines endoplasmic reticulum stress

Milena Vitale[1†], Anush Bakunts[1†], Andrea Orsi[1,2], Federica Lari[1,3], Laura Tadè[1], Alberto Danieli[1], Claudia Rato[4], Caterina Valetti[5], Roberto Sitia[1,2], Andrea Raimondi[6], John C Christianson[3‡], Eelco van Anken[1,2]*

[1]Division of Genetics and Cell Biology, San Raffaele Scientific Institute, Milan, Italy; [2]Università Vita-Salute San Raffaele, Milan, Italy; [3]Ludwig Institute for Cancer Research, University of Oxford, Oxford, United Kingdom; [4]Cambridge Institute for Medical Research, University of Cambridge, Cambridge, United Kingdom; [5]Department of Experimental Medicine, University of Genova, Genova, Italy; [6]Experimental Imaging Center, San Raffaele Scientific Institute, Milan, Italy

*For correspondence: evananken@mac.com

[†]These authors contributed equally to this work

Present address: [‡]NDORMS, University of Oxford, Oxford, United Kingdom

Competing interests: The authors declare that no competing interests exist.

**Abstract** How endoplasmic reticulum (ER) stress leads to cytotoxicity is ill-defined. Previously we showed that HeLa cells readjust homeostasis upon proteostatically driven ER stress, triggered by inducible bulk expression of secretory immunoglobulin M heavy chain ($\mu_s$) thanks to the unfolded protein response (UPR; Bakunts et al., 2017). Here we show that conditions that prevent that an excess of the ER resident chaperone (and UPR target gene) BiP over $\mu_s$ is restored lead to $\mu_s$-driven proteotoxicity, i.e. abrogation of HRD1-mediated ER-associated degradation (ERAD), or of the UPR, in particular the ATF6$\alpha$ branch. Such conditions are tolerated instead upon removal of the BiP-sequestering first constant domain ($C_H1$) from $\mu_s$. Thus, our data define proteostatic ER stress to be a specific consequence of inadequate BiP availability, which both the UPR and ERAD redeem.
DOI: https://doi.org/10.7554/eLife.41168.001

## Introduction

It is well-established that accumulation of unfolded proteins in the endoplasmic reticulum (ER)—a condition referred to as ER stress—activates the unfolded protein response (UPR), which, in turn, mitigates the stress, most notably through enhancing the ER chaperone content to boost the protein folding capacity (*Walter and Ron, 2011*). What defines ER stress, and how ER stress may engender cytotoxicity, however, are poorly understood issues. Moreover, it is still debated what feature of ER stress activates the UPR. An important reason why these are still open questions is the wide-spread use of ER stress-eliciting drugs, such as tunicamycin (Tm), which inhibits N-glycosylation, or thapsi-gargin (Tg), which causes $Ca^{2+}$ efflux from the ER (*Walter and Ron, 2011*). These drugs have pleio-tropic effects and are inherently cytotoxic, hence obscuring important aspects of how ER homeostasis can be restored by virtue of the UPR or not. To overcome the shortcomings of ER stress-eliciting drugs, we recently have developed a HeLa cell-based model for proteostatically driven ER stress (*Bakunts et al., 2017*). Inducible overexpression of the IgM subunits $\mu_s$ and the $\lambda$ light chain, in stoichiometric amounts, leads to bulk secretion of IgM with little if any UPR activation. In the absence of $\lambda$, however, $\mu_s$ is retained in the ER, and maximally activates the three main UPR branches, governed by IRE1$\alpha$, PERK, and respectively, ATF6$\alpha$. Yet, the cells successfully adapt to the proteostatic insult by expanding the ER both in size and in chaperone content, such that cell viability and growth are unaffected in the process, and UPR signaling subsides to a submaximal ampli-tude once homeostasis is restored (*Bakunts et al., 2017*).

The ER resident chaperone BiP stands out in the course of the adaptation to $\mu_s$ expression in two ways. First, ER stress sensing and UPR signaling occur in a $\mu_s$/BiP ratiometric fashion, that is the amplitude of UPR signaling is maximal when $\mu_s$ levels eclipse those of BiP, which is sequestered through binding to $\mu_s$, while UPR signaling subsides to submaximal output when an excess of BiP over $\mu_s$ is restored (*Bakunts et al., 2017*). ER homeostatic readjustment is due to the UPR, since BiP is a key UPR target gene (*Walter and Ron, 2011*). Second, ER homeostatic readjustment to $\mu_s$ expression causes a ~10-fold increase of BiP levels overall, which entails that BiP shifts from about one tenth to about one third of the total protein mass in the ER, such that BiP is the only chaperone in the ER of which the levels outmatch those of $\mu_s$ (*Bakunts et al., 2017*).

The two main models that have been proposed for UPR activation are that it entails i) dissociation of BiP from the lumenal domains of the main ER stress sensors, IRE1$\alpha$, PERK (*Bertolotti et al., 2000*) and ATF6$\alpha$ (*Shen et al., 2002*), and ii) direct binding of unfolded proteins (*Gardner and Walter, 2011*; *Karagöz et al., 2017*), including the Ig heavy chain $C_H1$ domain (*Karagöz et al., 2017*), to these sensors. Based on insights obtained from $\mu_s$-driven ER stress, we argue that these two UPR activation models are not mutually exclusive. Rather, the two models are complementary and should be unified, since in a three-way competition between UPR sensors, BiP, and an ER client protein ($\mu_s$) for binding one another, the ratio of UPR sensors bound to the client versus those bound to BiP most robustly report on the client/BiP ratio, to which indeed the UPR signaling amplitude correlates (*Bakunts et al., 2017*).

HeLa cells tolerate genetic ablation of the main three UPR transducers, but expression of $\mu_s$ in the context of UPR-ablated cells causes synthetic lethality through apoptosis, underscoring the key role the UPR has in restoring ER homeostasis (*Bakunts et al., 2017*). In this study we exploited this synthetic lethality to define how ER stress becomes proteotoxic.

## Results

### IRE1$\alpha$ and PERK are expendable, but ATF6$\alpha$ is key for $\mu_s$-provoked ER homeostatic readjustment

To investigate in detail how the UPR sustains ER homeostatic readjustment to bulk $\mu_s$ expression, we exploited cells in which IRE1$\alpha$ was deleted and PERK and ATF6$\alpha$ were silenced with good efficiency (*Bakunts et al., 2017*), either individually or in combinations. Surprisingly, ablation of IRE1$\alpha$ and PERK (either individually or in combination) had negligible effects on viability and growth of $\mu_s$-expressing cells, (*Figure 1A,B*), or on ATF6$\alpha$ activation (*Figure 1C*). Thus, IRE1$\alpha$ and PERK are dispensable for restoring ER homeostasis upon bulk $\mu_s$ expression, and ER stress levels are not enhanced in their absence (although there is some ATF6$\alpha$ activation already under basal conditions when IRE1$\alpha$ and PERK are ablated; *Figure 1C*). Conversely, silencing of ATF6$\alpha$ alone caused reduced growth and/or viability of $\mu_s$-expressing cells (*Figure 1A,B*), implying that ER homeostasis was not (fully) restored.

When $\mu_s$ is expressed for 3 days in wild-type cells, ER homeostasis is restored, and, consequently, IRE1$\alpha$ and PERK signaling subsides to submaximal output (*Bakunts et al., 2017*). In ATF6$\alpha$-silenced cells, conversely, ER homeostasis is not restored, and, accordingly, signaling through the PERK and IRE1$\alpha$ pathways remained persistently high (*Figure 1C*); that is levels of CHOP, a key downstream effector of PERK (*Harding et al., 2000*), were increased, and IRE1$\alpha$-mediated XBP1 mRNA splicing (*Calfon et al., 2002*) was enhanced, as was evident from the increased prominence of the higher mobility band, corresponding to the RT-PCR product of the $XBP1^S$ transcript from which the intron has been removed (*Calfon et al., 2002*). Ablation of ATF6$\alpha$ in combination with ablation of IRE1$\alpha$ and/or PERK caused apoptosis (*Bakunts et al., 2017*) and, consequently, abrogated viability of $\mu_s$-expressing cells (*Figure 1A,B*). We concluded that accumulation of $\mu_s$ in the ER per se confers proteotoxicity when the UPR is dysfunctional, and that the UPR counteracts this proteotoxicity, in particular through the ATF6$\alpha$ branch.

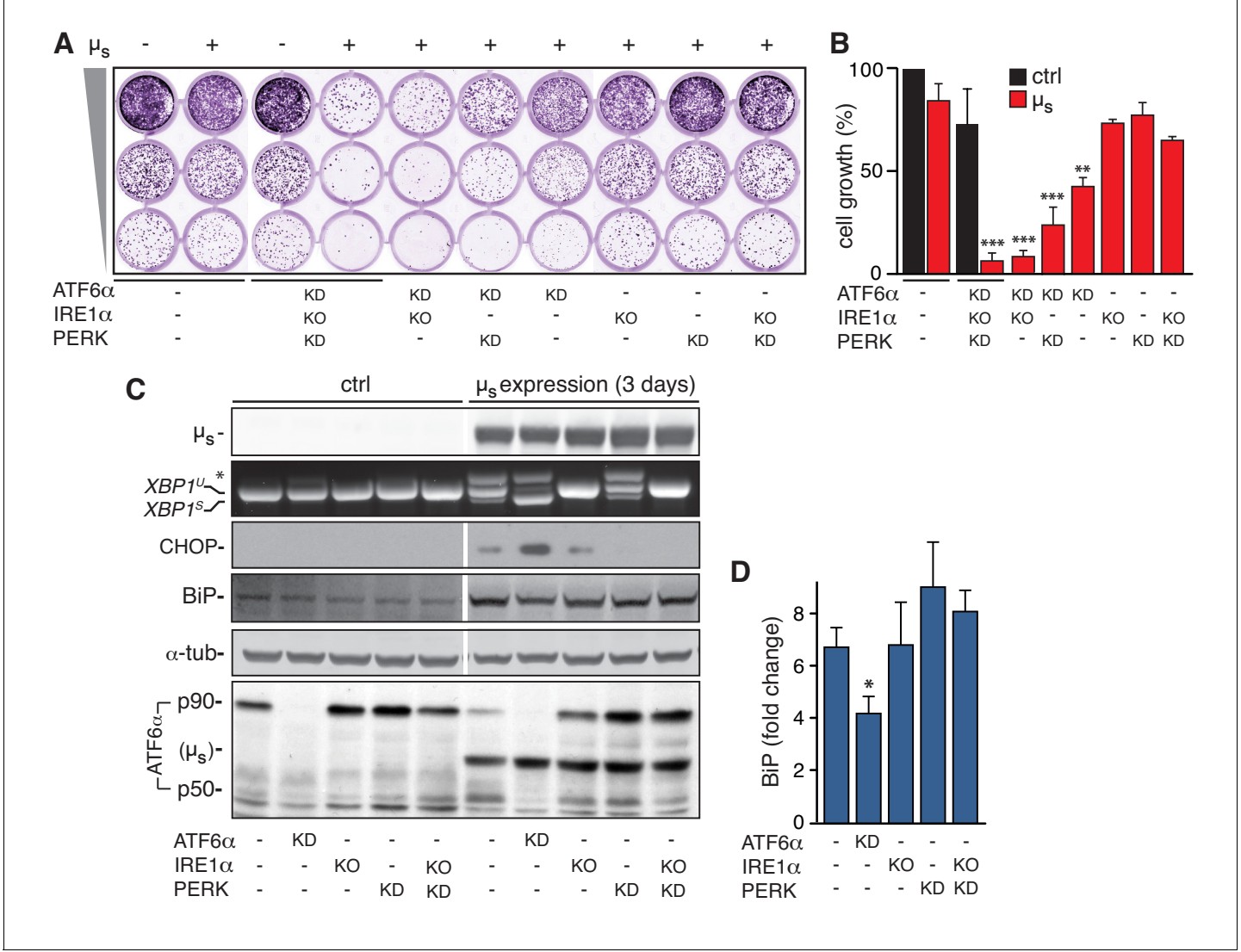

**Figure 1.** ATF6α is essential but IRE1α and PERK are dispensable for restoring ER homeostasis upon μ$_s$ expression. (**A–D**) In HeLa-μ$_s$ cells, IRE1α was deleted (KO), and ATF6α and PERK were silenced (KD) either alone or in combination, or not (-), as indicated. (**A**) Cells were seeded upon 1:5 serial dilution into 24-well plates, and treated with 0.5 nM mifepristone (Mif) to induce expression of μ$_s$ where indicated (+). After 7 days of growth, cells were fixed and stained with crystal violet. (**B**) Staining in (**A**) was quantitated as a measure for cell growth. Mean and s.e.m. are shown in a bar graph; n = 2. (**C**) Expression of μ$_s$ was induced for 0 or 3 days. Immunoblotting of lysates from cells that were sufficiently viable upon the insult for analysis revealed levels of μ$_s$, BiP, CHOP, α-tubulin, and ATF6α processing (i.e. release of the p50 cleavage product from the p90 precursor); cross-reaction of the secondary antibody against anti-ATF6α with μ$_s$ is denoted (μ$_s$). RT-PCR fragments corresponding to spliced (*XBP$^S$*) and unspliced (*XBP$^U$*) were separated on gel. A hybrid product that is formed during the PCR reaction is denoted by an asterisk. (**D**) BiP levels in (**C**) were quantitated and expressed as fold change upon μ$_s$ expression compared to untreated cells. Mean and s.e.m. are shown in a bar graph; n=2-5. Statistical significance of differences in growth (**B**), or in expression levels (**D**), was tested by ANOVA (*p ≤ 0.05; **p ≤ 0.01; ***p ≤ 0.001).

DOI: https://doi.org/10.7554/eLife.41168.002

The following source data is available for figure 1:

**Source data 1.**
DOI: https://doi.org/10.7554/eLife.41168.003

## IRE1α and PERK are expendable, but ATF6α is key for ER expansion in response to μ$_s$ expression

Despite the persistently maximal signaling through the PERK and IRE1α pathways upon μ$_s$ expression in ATF6α-silenced cells (*Figure 1C,D*), upregulation of BiP was compromised (*Figure 1C,D*;

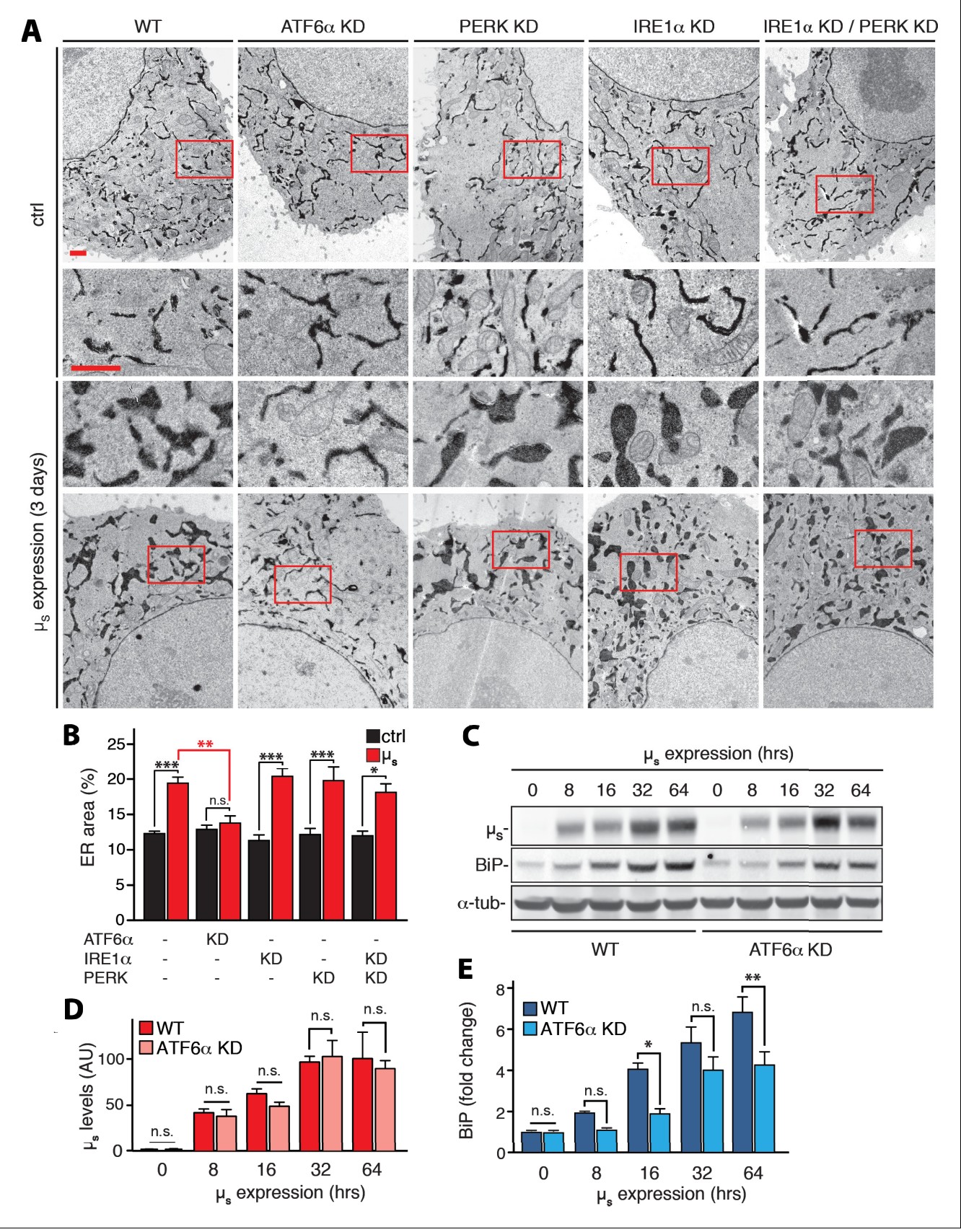

**Figure 2.** ATF6α is essential but IRE1α and PERK are dispensable for upregulation of ER chaperones and ER expansion in response to μ$_s$ expression. (A,B) HeLa-μ$_s$ cells in which UPR transducers were ablated by silencing alone or in combination, or not (WT), as indicated, were induced with 0.5 nM Mif to express μ$_s$ for 3 days or not. The cells harbor APEX-KDEL, a modified version of pea peroxidase that is targeted to the ER, and that catalyzes polymerization of 3,3'-diaminobenzidine tetrahydrochloride (DAB) upon treatment with $H_2O_2$ to obtain DAB precipitates (dark), revealing the extent of the ER in electron micrographs. Boxed areas are shown by 3-fold magnification; scale bars represent 1 μm (A). The extent of ER expansion was assessed as described (*Bakunts et al., 2017*), and the percentage of the area within the cytoplasm corresponding to ER was determined and depicted in bar graphs (B). Mean and s.e.m. are shown, n = 10–20. (C–E) Cells were induced to express μ$_s$ for the indicated times. Levels of μ$_s$ (D) and BiP (E) were quantitated from (C), and replicate experiments. (D) Levels in WT of μ$_s$ at 64 hr were set at 100 that was scaled to levels of BiP in WT at 64 hr such as to reflect a ratio of μ$_s$ to BiP of 2:3, that is an estimate for this ratio at day three based on earlier quantitations that we have described (*Bakunts et al., 2017*). Mean and s.e.m. are shown in bar graphs; n = 2–5. Statistical significance in the extent of ER areas in the electron micrographs between μ$_s$-expressing or non-expressing cells (black), or between μ$_s$-expressing WT or ATF6α ablated cells (red) (B), or in expression levels (D,E) was tested by ANOVA (n.s., not significant; *$p \leq 0.05$; **$p \leq 0.01$; ***$p \leq 0.001$).
DOI: https://doi.org/10.7554/eLife.41168.004

The following source data and figure supplement are available for figure 2:

**Source data 1.**
DOI: https://doi.org/10.7554/eLife.41168.005
**Figure supplement 1.** ATF6α ablation compromises ER chaperone upregulation upon μ$_s$ expression.
DOI: https://doi.org/10.7554/eLife.41168.006

*Figure 2C,E*), while upregulation of two other ER chaperones, PDI, and GRP94 was abolished (*Figure 2—figure supplement 1*), which confirms that also these ER chaperones are prominent ATF6α targets (*Bommiasamy et al., 2009*). ATF6α silencing did not affect accumulation of μ$_s$ (*Figure 2C, D*), however, and the ER did not expand (*Figure 2A, B*), in accordance with the compromised upregulation of ER chaperones. Conversely, ER expansion (*Figure 2A, B*), and BiP upregulation (*Figure 1C, D*) upon μ$_s$ expression was not compromised in PERK– and/or IRE1α–ablated cells. Thus, the ATF6α branch of the UPR is the main if not sole driver of ER expansion in response to μ$_s$ expression.

## ER stress and ensuing cytotoxicity levels correlate with the extent of μ$_s$ being chaperoned

Since the UPR induces expression of ER resident chaperones, we surmised that μ$_s$-driven ER stress becomes cytotoxic when the UPR is compromised, in particular upon ATF6α ablation, due to 'under-chaperoning' of μ$_s$. Proteins that undergo folding tend to aggregate in absence of sufficient folding assistance. Upon ablation of IRE1α and ATF6α, μ$_s$ indeed formed extensively disulfide-linked high molecular weight species that partitioned into a NP40-insoluble fraction, indicative of aggregation (*Mattioli et al., 2006*; *Valetti et al., 1991*)—with the single ablations showing intermediate phenotypes—(*Figure 3A*).

Under basal conditions, a significant proportion of BiP readily converts into an inactive, AMPylated state upon a three-hour block of protein synthesis with cycloheximide (CHX) (*Figure 3B*), which indicates that BiP gets to be dismissed from its chaperoning duties once its regular clients have had sufficient time to complete their folding, as has been reported before (*Preissler et al., 2015*). Conversely, in μ$_s$-expressing cells no AMPylation occurred upon CHX treatment at any time upon the onset of μ$_s$ expression (*Figure 3B*), suggesting that the vast majority of BiP is permanently engaged in chaperoning μ$_s$ even though the BiP pool is expanding massively in response to μ$_s$ expression (*Bakunts et al., 2017*).

As BiP stands out as a key chaperone for orphan μ$_s$, we reasoned that the level of BiP at basal conditions is a key determinant for μ$_s$-driven ER stress susceptibility. To test this idea, we created a derivative of the HeLa-μ$_s$ cell line with an integrated copy of the hamster HSPA5 gene that encodes BiP under control of doxycycline (Dox). The induction of μ$_s$ with Mif leads to it being the most abundantly transcribed gene (*Bakunts et al., 2017*) in the cells and concomitant induction of other transgenes would lead to competition for the transcription and/or translation machineries (not shown), thereby mitigating μ$_s$ expression and, hence, μ$_s$-driven ER stress by default. We therefore decided to pre-emptively enhance BiP levels with Dox at least ~10 fold prior to induction of μ$_s$ expression (*Figure 3C*). Even though exogenously driven BiP transcription ceased after that, exogenous (hamster) BiP levels remained high for a prolonged time (*Figure 3C*).

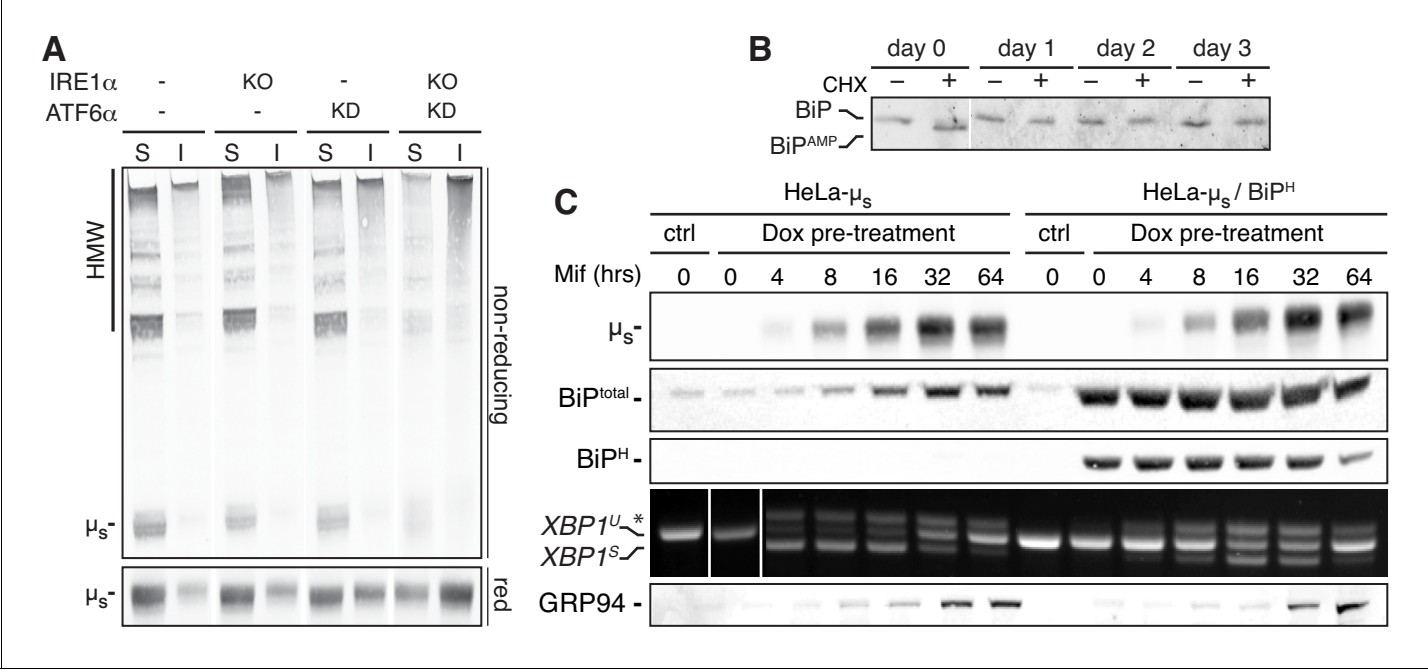

**Figure 3.** ER stress correlates with the extent of ER chaperones being engaged and becomes cytotoxic when their capacity is exceeded. (A) HeLa-$\mu_s$ cells, in which IRE1α (KO) and/or ATF6α (KD) was ablated, or not (-), as indicated, were induced with 0.5 nM Mif to express $\mu_s$ for 24 hr. Samples were lysed in NP40 and equivalent amounts of soluble (S) and insoluble (I) fractions resolved under reducing (red) or non-reducing conditions, blotted and decorated with anti-$\mu_s$. (B) HeLa-$\mu_s$ cells were induced with 0.5 nM Mif to express $\mu_s$ for the indicated times and treated with or without 100 µg/ml CHX for 3 hr before harvesting. Samples were analyzed by iso-electric focusing (IEF) to separate AMPylated (BiP$^{AMP}$) from non-AMPylated BiP, which were detected by immunoblotting, as described (*Preissler et al., 2015*). To allow a better comparison between samples, considering the upregulation of BiP upon $\mu_s$ expression, approximately 15 µg of lysates were loaded for the 0 day samples, while only 2.5 µg were loaded for the other days. (C) HeLa-$\mu_s$-derived cells, harboring Dox-inducible hamster BiP (HeLa-$\mu_s$/BiP$^H$), were treated for 2 days with 50 nM Dox to induce hamster BiP expression, while WT HeLa-$\mu_s$ cells were mock-treated with 50 nM Dox, before both cell lines were induced with 0.5 nM Mif to express $\mu_s$ for the indicated times. Immunoblotting of lysates revealed levels of $\mu_s$, total BiP, hamster BiP, and GRP94. *XBP1* mRNA splicing was assessed as in *Figure 1C*.
DOI: https://doi.org/10.7554/eLife.41168.007

In line with the notion that ER stress sensing in the HeLa-$\mu_s$ model occurs in a $\mu_s$/BiP ratiometric fashion (*Bakunts et al., 2017*), and in line with earlier reports that BiP overexpression dampens UPR activation (*Bertolotti et al., 2000*), *XBP1* mRNA splicing and upregulation of the UPR target GRP94 occurred with a delay when BiP levels were exogenously boosted as compared to when BiP was at endogenous levels, in spite of the similar extent and kinetics of $\mu_s$ accumulation (*Figure 3C*). Altogether the HeLa-$\mu_s$ model thus provides further support that sensing of ER stress correlates with the extent of the folding machinery being engaged in chaperoning its clients, and that BiP sequestration by client proteins appears to serve as the main proxy for that.

## Turnover of $\mu_s$ as afforded by ERAD is remarkably robust

While $\mu_s$ levels increase, and the ER expands (~3–4 fold compared to basal levels), as wild-type cells are still adapting to the proteostatic insult, there is no further build-up of $\mu_s$ levels and ER expansion after ~2–3 days once homeostasis is restored (*Bakunts et al., 2017*), which implies that at that stage the influx of $\mu_s$ molecules into the ER must be matched by countermeasures. Translational attenuation through PERK activation can alleviate the burden on the ER folding machinery by diminishing the input of nascent clients entering the ER lumen (*Harding et al., 1999*). Yet, we ruled out that PERK-driven translational attenuation was a key determinant for ER homeostatic readjustment in the HeLa-$\mu_s$ model, considering that PERK ablation hardly impeded cell growth upon $\mu_s$ expression (*Figure 1A,B*). Accordingly, there was only a marginal reduction in overall protein synthesis (being at the lowest ~80% of that before induction) that was moreover transient (i.e. only manifest during the first 16 hr of $\mu_s$-expression) (*Figure 4—figure supplement 1*). Following the same reasoning, we

also ruled out that regulated IRE1α-dependent decay (RIDD) (*Hollien and Weissman, 2006*; *Hollien et al., 2009*) of mRNAs that encode ER client proteins (and thereby limiting their influx into the ER) is important for homeostatic readjustment upon $\mu_s$ expression, since ablation of IRE1α had negligible impact on cell growth (*Figure 1A,B*). However, $\mu_s$ is a target of ERAD, as has been shown in plasma cells (*Fagioli and Sitia, 2001*), and which is shown here for the HeLa-$\mu_s$ cell model, since the proteasomal inhibitor MG132 to a large extent stabilizes $\mu_s$ levels in pulse-chase assays (*Figure 4A,B*). Since $\mu_s$ is glycosylated, it is subject to mannose trimming (*Aebi et al., 2010*), which is a key step in delivering $\mu_s$ to the retro-translocation machinery that shuttles it to the cytosol for proteasomal degradation (*Fagioli and Sitia, 2001*). Accordingly, the ER mannosidase I inhibitor kifunensine (Kif) stabilized $\mu_s$ in a similar manner as MG132 (*Figure 4A,B*).

Interestingly, while $\mu_s$ levels built up steadily in the ER with time, ERAD kinetics hardly changed (i.e. the half-life ($t_{1/2}$) of $\mu_s$ was remarkably constant), which implies that ERAD prowess kept pace with the accumulating load of $\mu_s$ (*Figure 4C*). ERAD components are UPR target genes (*Walter and Ron, 2011*), and indeed various major ERAD components (HRD1, SEL1L, Ube2j1, HERP, and OS-9), which we previously failed to detect by proteomics (*Bakunts et al., 2017*), were upregulated upon $\mu_s$ expression (*Figure 4—figure supplement 2*), Yet, their upregulation apparently serves at most to maintain rather than to reinvigorate ERAD kinetics of the accumulating $\mu_s$ load. In fact, ERAD kinetics of $\mu_s$ were not markedly affected by ablation of ATF6α (*Figure 4D,E*), in line with the finding that intracellular $\mu_s$ accumulation was not aggravated upon ATF6α ablation (*Figure 2C,D*). Thus, ER homeostatic failure upon ATF6α ablation is not due to compromised ERAD. Apparently, the upkeep of ERAD is robust in HeLa-$\mu_s$ cells, since we can also rule out that IRE1α and/or PERK are essential for maintaining sufficient ERAD capacity, as their ablation hardly caused any growth impairment of $\mu_s$-expressing cells (*Figure 1A,B*), unlike when ERAD is inhibited—see below.

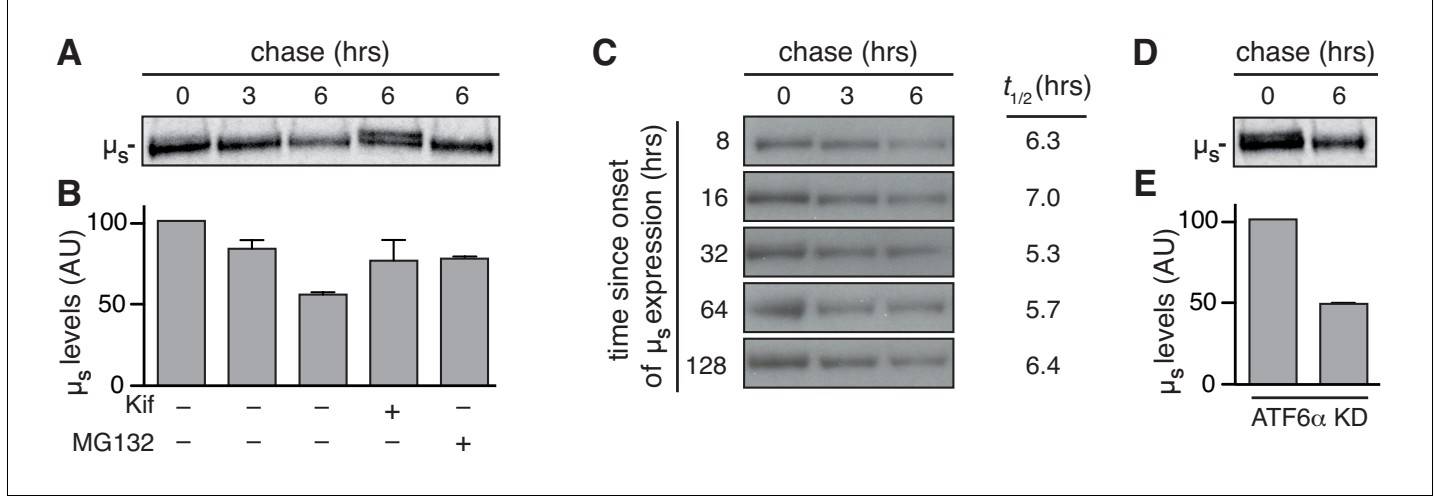

**Figure 4.** ERAD accounts for disposal of $\mu_s$ in a robust manner. HeLa-$\mu_s$ cells, in which ATF6α was ablated (**D,E**), as indicated, or not (**A-C**) were pulse labeled for 10 min and chased with excess unlabeled cysteine and methionine for the indicated times after 24 hr (**A,B,D,E**) or at various times (**C**), as indicated, after induction of $\mu_s$ expression with 0.5 nM Mif, in the absence (**A,C,D**) or presence (**A**) of 10 μM MG132 or 30 μM Kif, as indicated (+). Signals were quantitated and the signal after 0 hr chase was set at 100; mean and s.e.m. are shown in bar graphs (**B,E**); linear fitting of the quantitations of (**C**) were used to calculate the $t_{1/2}$ of $\mu_s$ at various time points after induction of its expression; see column on the right of panel.
DOI: https://doi.org/10.7554/eLife.41168.008

The following source data and figure supplements are available for figure 4:

**Source data 1.**
DOI: https://doi.org/10.7554/eLife.41168.012

**Figure supplement 1.** Translation is transiently attenuated upon $\mu_s$ overexpression to a marginal extent.
DOI: https://doi.org/10.7554/eLife.41168.009

**Figure supplement 1—source data 1.**
DOI: https://doi.org/10.7554/eLife.41168.010

**Figure supplement 2.** Levels of various ERAD components are induced in response to $\mu_s$ expression.
DOI: https://doi.org/10.7554/eLife.41168.011

## Disposal of $\mu_s$ through HRD1 complex-mediated ERAD is key for homeostatic readjustment

While prolonged proteasomal inhibition in itself is cytotoxic, blocking ERAD of glycoproteins with Kif per se did not affect cell viability (*Figure 5A*), and did not activate the UPR either (*Figure 5—figure supplement 1*). We reasoned that ERAD would be important, however, to hold bulk accumulation of $\mu_s$ in check. Indeed, viability was compromised in Kif-treated $\mu_s$-expressing cells (*Figure 5A*). Key ERAD components are the E3 ligase HRD1 and its partner SEL1L (*Olzmann et al., 2013*), which have previously been shown to mediate ERAD of $\mu_s$ (*Cattaneo et al., 2008*). Indeed, ablation of HRD1 and, to a lesser extent, of SEL1L was synthetically lethal in HeLa-$\mu_s$ cells upon $\mu_s$ expression (*Figure 5A*).

HRD1 and SEL1L cooperate to target ERAD substrates back across the ER membrane to the cytosol, where substrates are ubiquitinated, deglycosylated by N-glycanase, and, ultimately, degraded by the proteasome (*Olzmann et al., 2013*). Accordingly, $\mu_s$ was stabilized in HRD1 KO or SEL1L KD cells, similarly as upon Kif treatment of WT cells, while in ERAD-competent WT cells $\mu_s$ was degraded upon CHX treatment (*Figure 5B,C*). Proteasomal inhibition with MG132 stabilized $\mu_s$ in WT cells, and the appearance of a deglycosylated form of $\mu_s$ confirmed that, at least of fraction of $\mu_s$ was retrotranslocated to the cytosol, and accessible to N-glycanase. Interestingly, in HRD1 KO or SEL1L KD cells no deglycosylated form of $\mu_s$ appeared, indicating that disposal of $\mu_s$ was blocked at (or prior to) the retrotranslocation step (*Figure 5C*).

There appear to be more than 25 other E3 ligases that localize at the ER membrane next to HRD1 (*Neutzner et al., 2011*; *Kaneko et al., 2016*), but, curiously, none of these can compensate for the loss of HRD1. Furthermore, treatment with the autophagy inhibitor Bafilomycin A1 (BafA1) did not lead to any stabilization of $\mu_s$ (*Figure 5B*). Thus, HRD1-mediated ERAD is the main if not exclusive disposal mechanism that is essential for ER homeostatic readjustment in the HeLa-$\mu_s$ model, even though autophagy has been reported to curtail IgM production and ER expansion in plasma cells (*Pengo et al., 2013*).

The synthetic lethality that ensues once ERAD is compromised in the HeLa-$\mu_s$ model offered a powerful tool to define which factors are crucial to act in conjunction with HRD1 and SEL1L in the disposal of $\mu_s$. To that end, we ablated several candidate HRD1 partners by CRISPR/Cas9 (but without clonal selection; that is without necessarily reaching fully penetrant phenotypes). In this initial survey, we witnessed that cell viability upon $\mu_s$-expression was compromised, and that $\mu_s$ was stabilized in a CHX chase by ablation of HERP, Ube2j1, and Derlin2 to significant extents (*Figure 5—figure supplement 2*). However, ablation of OS-9 or of XTP3-B only mildly affected $\mu_s$-expressing cells. These two lectins indeed have been shown previously to be interchangeable, as they capture soluble ERAD substrates upon mannose trimming of their glycans before handing over these substrates to SEL1L (*Bernasconi et al., 2010*; *van der Goot et al., 2018*).

In sedimentation gradients HRD1, SEL1L, and Derlin2 shifted towards heavier fractions upon $\mu_s$ expression, while the redundant ERAD factor OS-9 did not (*Figure 5D*). These findings indicate that disposal of $\mu_s$ is effectuated through assembly of higher-order ERAD-mediating complexes with, at the least, HRD1, SEL1L, and Derlin2 at their core. These complexes nucleate around HRD1 as its ablation abrogated their formation (*Figure 5D*).

## Homeostatic failure upon ERAD inhibition coincides with $\mu_s$ levels outpacing BiP upregulation

When ERAD is functional, an excess of BiP over $\mu_s$ is restored upon 3 days of $\mu_s$ expression. The BiP:$\mu_s$ stoichiometry is then ~3:2, as estimated from a combination of quantitative immunoblotting and proteomics techniques (*Bakunts et al., 2017*). As soon as BiP levels are in excess again, UPR signaling subsides to submaximal output, and ER homeostatic readjustment to $\mu_s$ expression is successful (*Bakunts et al., 2017*). The loss of viability in Kif-treated $\mu_s$-expressing cells (*Figure 5A*) indicated that ER homeostatic readjustment failed, and these cells indeed underwent apoptosis (*Figure 6A*). Homeostatic failure in Kif-treated $\mu_s$-expressing cells entailed that ER stress was unresolved, and accordingly, IRE1$\alpha$ and PERK chronically signaled at maximal levels (*Figure 6B*).

Chronic maximal UPR activation upon ERAD inhibition in $\mu_s$-expressing cells implied that induction of BiP expression was persistently at maximal levels. Nevertheless, the build-up of BiP levels (~2 fold further increase after 3 days), could not keep pace with the augmented accumulation of $\mu_s$ (~3

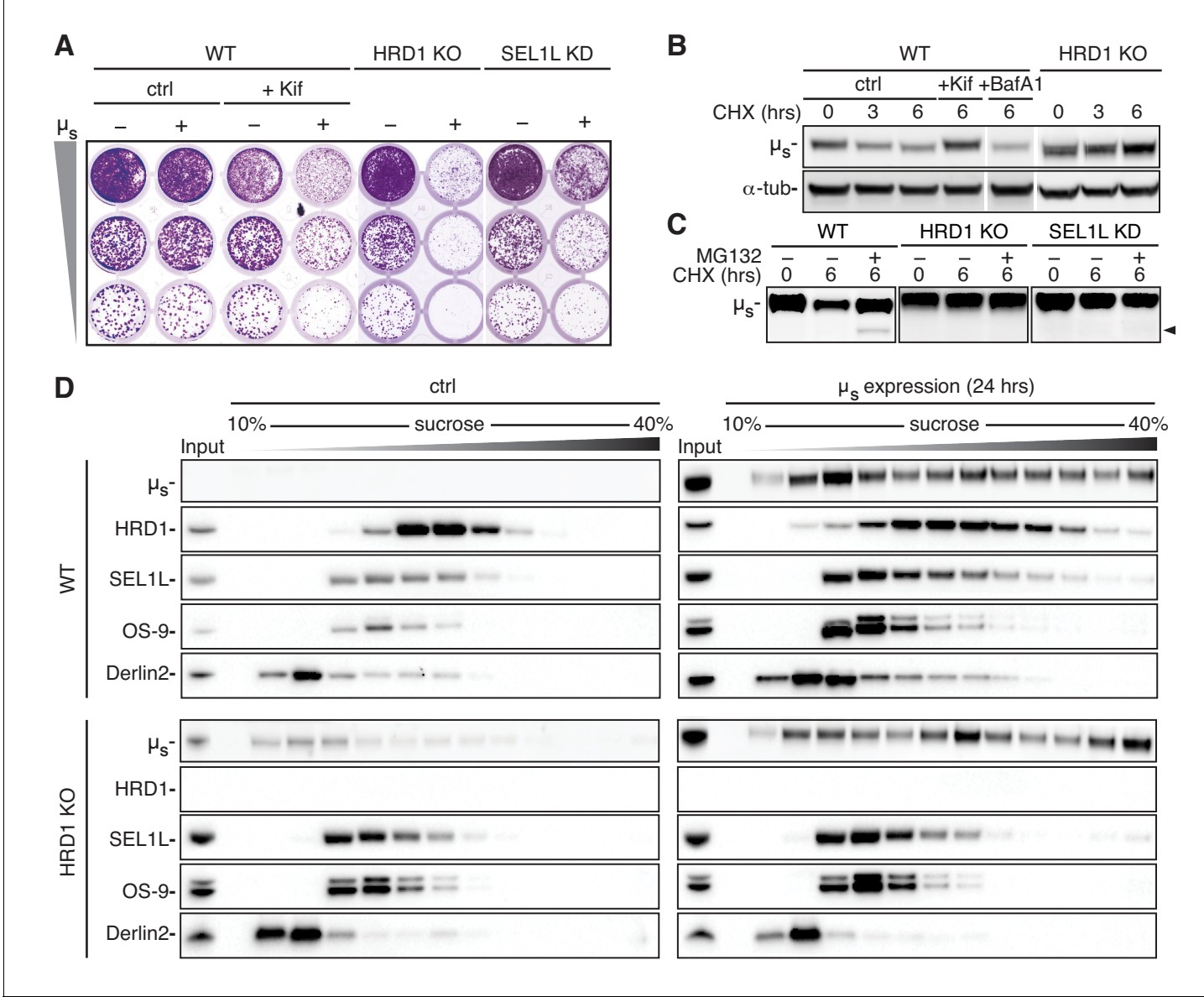

**Figure 5.** ERAD of $\mu_s$ is mediated through the HRD1 complex. (A) Growth assay as in *Figure 1A* of HeLa-$\mu_s$ cells, in which HRD1 was deleted (KO), SEL1L was silenced (KD), or not (WT). Cells were treated with 0.5 nM mifepristone (Mif) to induce expression of $\mu_s$ (+), or not (-), and WT cells were treated with Kif or not (ctrl), as indicated. (B,C) Immunoblots of $\mu_s$ harvested from WT, HRD1 KO (B,C), or SEL1L KD (C) HeLa-$\mu_s$ cells that were induced with 0.5 nM Mif to express $\mu_s$ for 4 hr and then treated for the indicated times with 100 µg/ml CHX either alone (B,C), in combination with 20 mM Kif, 100 nM BafA1, or not (ctrl) (B), or 10 µg/ml MG132 (C), as indicated. The arrowhead indicates the deglycosylated form of $\mu_s$. (D) HeLa-$\mu_s$ WT or HRD1 KO cells were induced with Mif (0.5 nM) for 24 hr to express $\mu_s$ or not (ctrl) , as indicated. Samples were lysed in 1% lauryl maltose neopentyl glycol (LMNG) and sedimented over a 10–40% sucrose gradient. Levels of $\mu_s$, HRD1, SEL1L, Derlin-2, and OS-9 were detected by immunoblotting. Note that in HRD1 KO cells leaky expression of $\mu_s$ becomes apparent due to the lack of ERAD. At low expression levels, however, $\mu_s$ does not form high molecular weight aggregates, indicative of the adequacy of the chaperoning machinery.

DOI: https://doi.org/10.7554/eLife.41168.013

The following figure supplements are available for figure 5:

**Figure supplement 1.** Inhibition of ERAD does not trigger the UPR under basal conditions.

DOI: https://doi.org/10.7554/eLife.41168.014

**Figure supplement 2 .** HERP, Ube2j1, and Derlin-2 are key for ERAD of $\mu_s$.

DOI: https://doi.org/10.7554/eLife.41168.015

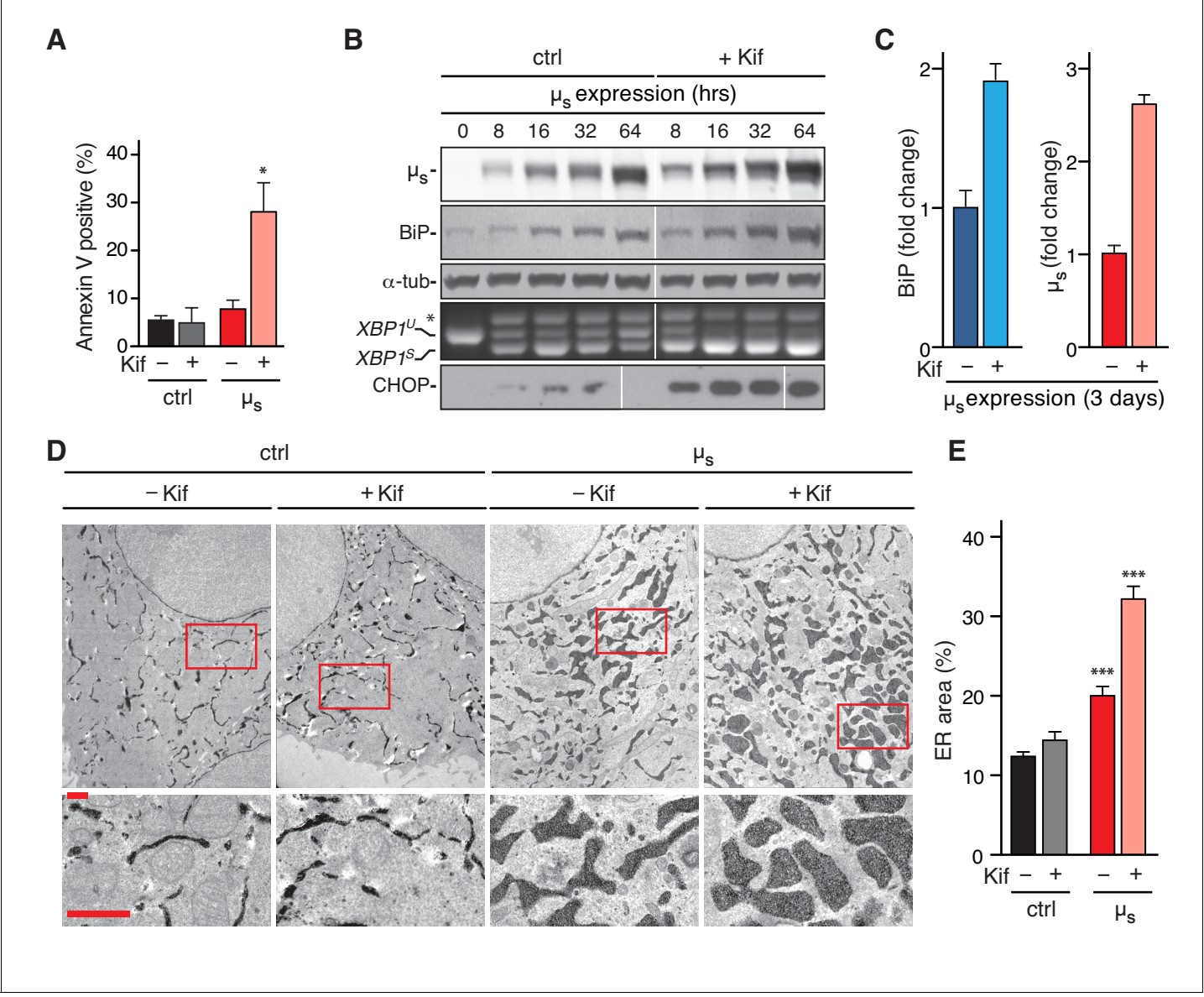

**Figure 6.** Abrogation of $\mu_s$ disposal through ERAD leads to BiP being permanently eclipsed, ER homeostatic failure, and apoptosis. (A,D,E) HeLa-$\mu_s$ cells, harboring APEX-KDEL (D,E) or not (A) were induced with ($\mu_s$) or without (ctrl) 0.5 nM Mif for 3 days in the presence or absence of 30 µM Kif. (A) Percentages of Annexin V positive cells were assessed by cytometric analysis. Mean and s.e.m. are shown in a bar graph, n = 2–4. (B,C) HeLa-$\mu_s$ cells were induced to express $\mu_s$ for various times as indicated (B) or for 3 days in the absence or presence of 30 µM Kif. (C) Levels of $\mu_s$, BiP, and $\alpha$-tubulin as well as activation of the IRE1$\alpha$ and PERK branches of the UPR were assessed as in (*Bakunts et al., 2017*). (B) Levels of BiP and $\mu_s$ were assessed by quantitative immunoblotting as described (*Bakunts et al., 2017*), and depicted in bar graphs as in *Figure 2D,E*, such that the $\mu_s$ levels in the absence of Kif were scaled to BiP levels at a ratio of 2:3. Levels in the presence of Kif are expressed as a fold change compared to levels in the absence of Kif; mean and s.e.m. are shown; n = 2. (D) In cells harboring APEX-KDEL the extent of ER expansion was assessed as in *Figure 2A*. Boxed areas are shown by 3-fold magnification; scale bars represent 1 µm. The percentage of the dark area within the cytoplasm corresponding to ER was determined and depicted in bar graphs (E), mean and s.e.m. are shown, n = 10. Statistical significance of differences in Annexin V staining (A), or the extent of ER occupying cytosolic area in the electron micrographs (E) were tested by ANOVA (*p≤0.05; ***p≤0.001).

DOI: https://doi.org/10.7554/eLife.41168.016

The following source data is available for figure 6:

**Source data 1.**

DOI: https://doi.org/10.7554/eLife.41168.017

fold further increase after 3 days) upon ERAD inhibition, such that $\mu_s$ reached levels in the ER that were at about a 1:1 stoichiometry with BiP (*Figure 6C*). Indeed, aggregation of $\mu_s$ increased when ERAD was defective, as judged by $\mu_s$ shifting more towards heavier fractions in HRD1 KO than in WT cells (*Figure 5D*). Thus, under those conditions the chaperoning machinery becomes limiting, similarly as upon ablation of IRE1α and/or ATF6α in ERAD-competent cells (*Figure 3A*).

We previously estimated the volume of the ER under basal conditions to be $(0.10–0.12)^{3/2} \approx 3–4\%$ of the cytoplasmic volume, and upon 3 days of $\mu_s$ expression to be $(0.18–0.20)^{3/2} \approx 7–8\%$ of the cytoplasmic volume, corresponding to a $\sim 2–3$ fold increase of ER volume (*Bakunts et al., 2017*). Upon ERAD inhibition with Kif the ER did not markedly expand in non-$\mu_s$-expressing cells. In $\mu_s$-expressing cells, instead, ERAD inhibition caused the area of ER staining within the cytoplasm to reach 30–35%, which on a rough estimate would account for $(0.3–0.35)^{3/2} \approx 17–20\%$ of the cytoplasmic volume, implying that the ER had expanded $\sim$6–7 fold since the onset of $\mu_s$-expression (*Figure 6D,E*).

We concluded that curtailing the $\mu_s$ load by ERAD is essential for the cells to cope with $\mu_s$ expression in bulk. ER homeostatic failure upon ERAD inhibition coincided with an inadequacy to raise BiP levels in sufficient excess over those of $\mu_s$ and, hence, with its 'under-chaperoning', in spite of the impressive BiP upregulation and ER expansion at large. Thus, in absence of ERAD, not only the chronic maximal UPR activation, but also the $\mu_s$–driven proteotoxicity appear to be due to BiP running short, similarly as when UPR signaling was compromised upon ablation of ATF6α (*Figure 1C,D*; *Figure 2C–E*).

## Sequestration of BiP is both necessary and sufficient for UPR activation and ER stress-provoked proteotoxicity

In plasma cells BiP stringently interacts with $\mu_s$ through the $C_H1$ domain, until it is displaced by the light chain (*Bole et al., 1986*; *Figure 7A*), which makes $\mu_s$ an unusual ER client. Evolutionary pressure against secretion of orphan $\mu_s$ (i.e. unaccompanied by the light chain) must have been extraordinarily high for obvious immunological reasons (*Anelli and van Anken, 2013*), which would explain the exceptionally strong affinity of the $C_H1$ domain for BiP, that is to let BiP mediate stringent ER retention of unpaired $\mu_s$. Thus, we reasoned that removal of the BiP binding $C_H1$ domain from $\mu_s$ (*Figure 7A*), would offer an ideal tool to validate whether limitations in BiP availability define both the amplitude of UPR activation as well as any proteotoxicity that would ensue from overexpression of ER client proteins. In line with our model, $\mu_s\Delta C_H1$ hardly activated the UPR, as shown for the IRE1α and PERK branches, despite being expressed at similar levels as $\mu_s$ wild-type (*Figure 7B*). Moreover, genetic ablation of the three UPR pathways failed to cause synthetic lethality in $\mu_s\Delta C_H1$-expressing cells (*Figure 7C*). Conversely, co-expression of $\mu_s\Delta C_H1$ with a chimeric protein consisting of the variable domain of λ fused with the $C_H1$ domain of $\mu_s$ ($V_L$-$C_H1$), which teams up with $\mu_s\Delta C_H1$ through interactions between $V_L$ with the variable domain of $\mu_s$ ($V_H$) (*Figure 7A*), restored UPR activation (*Figure 7B*) and synthetic lethality upon UPR ablation (*Figure 7C*). These findings corroborate that the BiP sequestering $C_H1$ domain of $\mu_s$ causes UPR activation, as well as proteotoxicity when reinforcement of BiP levels through the UPR is inadequate.

## Discussion

The fact that BiP plays a key role in regulating the UPR has been known for almost 20 years (*Bertolotti et al., 2000*). Overexpression of BiP dampens UPR activation (*Bertolotti et al., 2000*), which we confirm with the data presented here, while inactivating BiP with the AB5 subtilase cytotoxin acutely causes ER stress and UPR activation (*Paton et al., 2006*). Yet, by employment of a proteostatic stimulus with a single well-defined BiP binding module as the source of ER stress, we have provided here experimental evidence that defines both proteostatic ER stress, and the resulting activation of the UPR, to be the specific consequence of insufficient BiP availability. Both the UPR and ERAD redress the relative BiP shortage, and thus counteract that the proteostatic stress becomes proteotoxic. BiP indeed has been acclaimed as the master regulator of ER function (*Hendershot, 2004*), and various cytotoxic consequences may follow from the excessive sequestering of BiP by $\mu_s$ that precludes BiP from attending to its other functions. For instance, BiP closes off the translocon, and efflux of $Ca^{2+}$ from the ER into the cytosol through poorly gated translocons may already be sufficient to cause apoptosis (*Schäuble et al., 2012*).

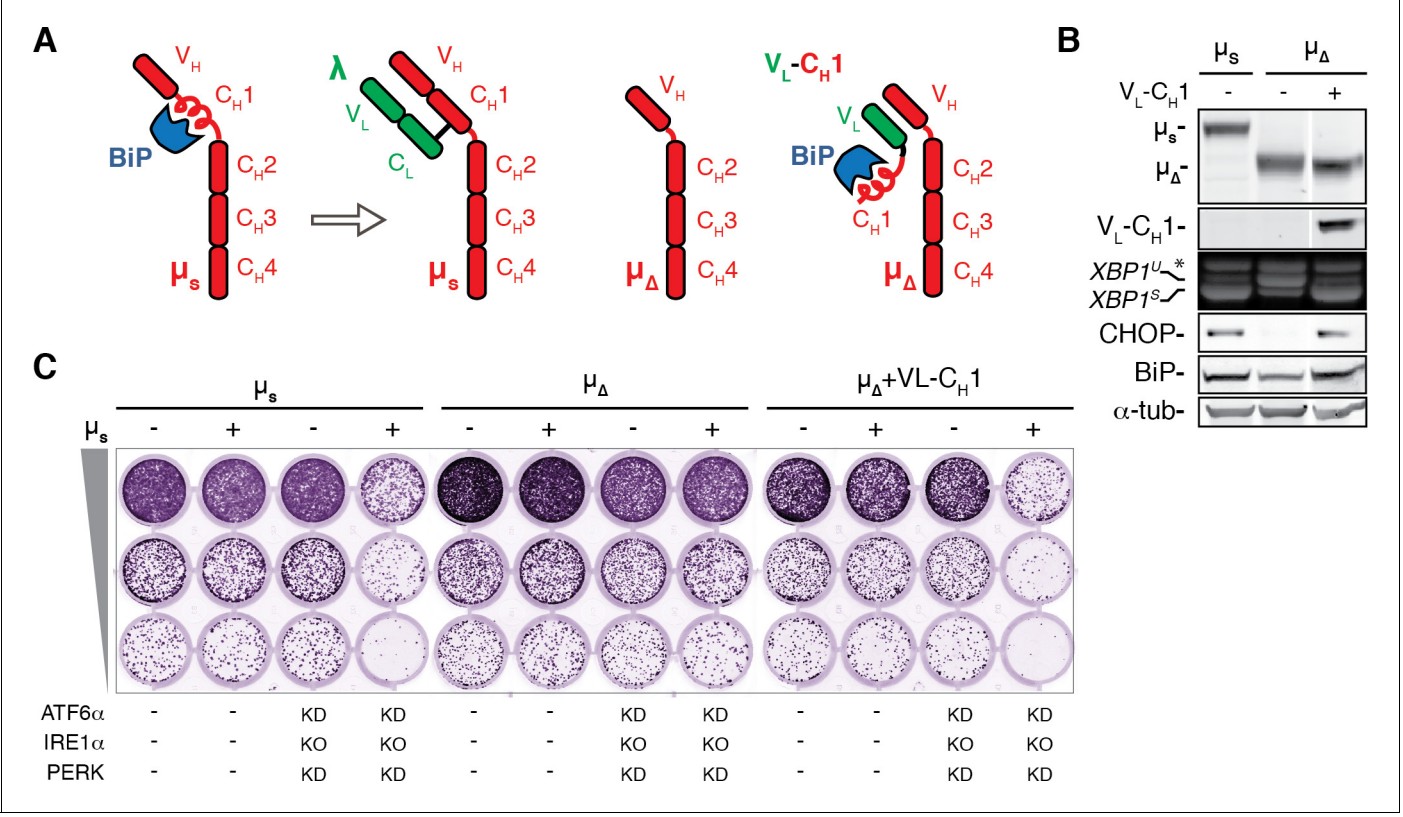

**Figure 7.** The BiP-sequestering $C_H1$ domain of $\mu_s$ is necessary and sufficient to cause UPR activation and proteotoxic ER stress in absence of the UPR. (A) Schematic representation of BiP associating with the $C_H1$ domain of $\mu_s$ until it is displaced by the light chain ($\lambda$). Deletion of the $C_H1$ domain ($\mu_\Delta$) abolishes BiP association, but through pairing of the $V_H$ and $V_L$ domains, the $C_H1$ domain can associate in trans by virtue of a synthetic chimeric $V_L$-$C_H1$ construct. (B) HeLa cells were induced for 24 hr with 0.5 nM Mif to express the transgenes $\mu_s$, $\mu_s\Delta C_H1$ ($\mu_\Delta$) alone or in conjunction with $V_L$-$C_H1$, as indicated. Immunoblotting of lysates revealed levels of $\mu_s$, $\mu_\Delta$, $V_L$-$C_H1$, BiP, CHOP, and $\alpha$-tubulin, as in *Figure 1A*. (C) Growth assay as in *Figure 1A* of HeLa cells inducibly expressing $\mu_s$, $\mu_s\Delta C_H1$ ($\mu_\Delta$) in conjunction with $V_L$-$C_H1$ or not, and in which the UPR was ablated (i.e. IRE1$\alpha$ was deleted (KO), and ATF6$\alpha$ and PERK were silenced in combination), or not, as indicated.

DOI: https://doi.org/10.7554/eLife.41168.018

Our data emphasize that proteotoxicity stemming from the accumulation of client proteins in the ER is not the result of UPR signaling, as often is assumed from the notion that the UPR can initiate pro-apoptotic pathways. Instead, the UPR foremost counteracts proteotoxicity by inducing the ER resident folding machinery (most in particular BiP). In light of our data, the capacity of the UPR to switch from cytoprotective to pro-apoptotic signaling may well have arisen in metazoans to pre-emptively eliminate cells in which restoration of ER homeostasis is unachievable, and, hence, cell death has become inevitable.

Perhaps surprisingly, our results furthermore highlight that PERK and IRE1$\alpha$ are dispensable for successful ER homeostatic readjustment to the $\mu_s$ stimulus in HeLa cells. Apparently, the PERK-mediated translational block, which is only transient, offers negligible advantage when cells face a sudden proteostatic insult that sequesters BiP (and/or the ER chaperone machinery at large) in a persistent manner. PERK-mediated translational attenuation instead may be required in particular to sustain episodic secretory activity, such as in $\beta$-cells of the pancreas. PERK KO mice indeed suffer mostly from degeneration of tissues with episodic secretory activity (*Zhang et al., 2002*). Similarly, requirements for the IRE1$\alpha$/XBP1 pathway seem to be tissue-specific. Both deletion of IRE1$\alpha$ (*Urano et al., 2000*; *Zhang et al., 2005*) and of XBP1 (*Reimold et al., 2000*) cause embryonic lethality, but XBP1 KO mice are rescued with an XBP1 transgene specifically expressed in the liver (*Lee et al., 2005*), while IRE1$\alpha$ KO mice are rescued when the placenta expresses IRE1$\alpha$ (*Iwawaki et al., 2009*), and the resulting rescued mice display relatively mild symptoms, that ishyperglycemia, hypoinsulinemia, and decreased antibody titers, despite the lack of IRE1$\alpha$ (*Iwawaki et al., 2010*). In line with our

findings, homeostatic readjustment to ER stress in most mammalian tissues seems to rely mainly on ATF6 proteins (*Wu et al., 2007*; *Yamamoto et al., 2007*), that is ATF6α, and its related ER stress sensor ATF6β. The ATF6α/β double KO confers embryonic lethality (*Yamamoto et al., 2007*). At present, it is unclear whether embryonic lethality of the ATF6α/β double KO can be rescued, for instance through enhancement of other UPR branches.

Finally, since proteotoxicity due to the accumulation of (mutant) proteins in the ER seems to play a key role in various types of disease (*Anelli and Sitia, 2010*; *Cao and Kaufman, 2014*), our insights may be of relevance for the design of drugs aimed at alleviating ER stress (*Hetz and Papa, 2018*), and hence proteotoxicity stemming from ER stress. We argue that pharmacological intervention against pathogenic ER stress foremost should promote a favorable ratio of BiP levels over those of its disease-causing client protein.

## Materials and methods

All assays were performed as described (*Bakunts et al., 2017*), except that in addition, along the same principles as described (*Bakunts et al., 2017*), the following cell lines were derived by clonal selection from either HeLa-$\mu_s$ or HeLa-MifON, as summarized in *Supplementary file 1*: HeLa-$\mu_s$ HRD1 KO, and HeLa-$\mu_s$-BiP$^H$, which inducibly (by Dox) expresses hamster BiP, HeLa-$\mu_s\Delta C_H1$, which inducibly (by Mif) expresses $\mu_s\Delta C_H1$, and HeLa-$\mu_s\Delta C_H1/V_L$-$C_H1$, which inducibly (by Mif) expresses $\mu_s\Delta C_H1$ in combination with $V_L$-$C_H1$. At least three independent clones of HeLa-$\mu_s$ HRD1 KO cells were tested in phenotypic assays to rule out off-target effects. CRISPR/Cas9-mediated depletion of HERP, Ube2j1, Derlin-2, OS-9, or XTP3-B in HeLa-$\mu_s$ was performed using single guide RNA (sgRNA) sequences (*Supplementary file 1*) cloned into the PX459 vector (Addgene #62988) and were used as puromycin-selected pools without clonal isolation. Cloning into PX459 was performed as described previously (*Ran et al., 2013*). sgRNA target sequences for Hrd1, HERP (*Schulz et al., 2017*), Derlin-2, Ube2j1 (*Ma et al., 2015*), OS-9, and XTP3-B (*van der Goot et al., 2018*) have been described previously. Silencing of SEL1L was obtained using ON-Target SMARTpool siRNA from Dharmacon.

The inducible hamster BiP, $\mu_s\Delta C_H1$ and $V_L$-$C_H1$ cassettes were created by standard molecular biology techniques from the cDNAs described in *Bakunts et al. (2017)*. The $C_H1$ domain (E140-P244) was deleted from $\mu_s$ in $\mu_s\Delta C_H1$. That same $C_H1$ domain was placed downstream of V127 of $\lambda$, replacing the $C_L$ domain, to create the chimeric $V_L$-$C_H1$ construct. A myc tag (EQKLISEEDL) was placed at the C-terminus of $V_L$-$C_H1$ for immunodetection purposes. Cells were routinely tested, that is on a monthly basis, to be mycoplasm-free by use of a standard diagnostic PCR. All cell lines in this study were ultimately derived from HeLa S3 cells, of which the genotype was confirmed by PCR single locus technology. Antibodies used in addition to those described before (*Bakunts et al., 2017*) are summarized in *Supplementary file 1*.

To separate NP-40 soluble from insoluble fractions, cells were washed and lysed in 0.2% NP-40, 50 mM Tris-HCl pH 7.5, 150 mM NaCl, 5 mM EDTA, 10 mM *N*-ethylmaleimide and a cocktail of protease inhibitors. The NP-40-insoluble fractions were separated from the soluble fractions by centrifugation at 3,400 g for 10 min and the insoluble pellets were solubilized in 1% SDS, 50 mM Tris-HCl pH 7.5, 10 mM NEM for 10 min at RT and sonicated on ice. For fractionation of ERAD complexes cells were lysed in 1% lauryl maltose neopentyl glycol (LMNG, Anatrace) containing buffer (50 mM Tris-HCl, pH7.4, 150 mM NaCl, 5 mM EDTA) and lysates were loaded onto 10–40% sucrose gradients also containing 1% LMNG, formed by following the manufacturers' instructions (Gradient Master, Biocomp). Sedimentation was achieved by centrifugation in a SW.41 swing bucket rotor (Beckman) at 39,000 rpm for 16 hr at 4°C. Thirteen fractions were collected from the top and proteins precipitated with TCA (trichloroacetic acid). Protein pellets were resuspended in Laemmli buffer containing DTT (10 mM), heated alongside 25 µg of the original lysates as input, at 56°C prior to separation by SDS-PAGE.

## Acknowledgements

All members of the ALEMBIC imaging facility and the Sitia and Van Anken labs are acknowledged for stimulating discussions and advice. We thank Drs Linda Hendershot, Ron Kopito, and Yihong Ye

for kindly sharing antibodies, and Drs Steffen Preissler and David Ron for their help with assessing the AMPylation state of BiP.

## Additional information

### Funding

| Funder | Grant reference number | Author |
| --- | --- | --- |
| Giovanni Armenise-Harvard Foundation | | Eelco van Anken |
| Ministero della Salute | RF-2011-02352852 | Eelco van Anken |
| Associazione Italiana per la Ricerca sul Cancro | MFAG 13584 | Eelco van Anken |
| Ministero della Salute | PE-2011-02352286 | Roberto Sitia Eelco van Anken |
| Associazione Italiana per la Ricerca sul Cancro | IG 2016-18824 | Roberto Sitia |
| Fondazione Telethon | GGP15059 | Roberto Sitia |
| Fondazione Cariplo | 2015-0591 | Roberto Sitia |
| Medical Research Council | MR/L001209/1 | John C Christianson |
| Ludwig Institute for Cancer Research | | John C Christianson |

The funders had no role in study design, data collection and interpretation, or the decision to submit the work for publication.

### Author contributions

Milena Vitale, Conceptualization, Data curation, Formal analysis, Supervision, Validation, Investigation, Visualization, Methodology, Writing—review and editing; Anush Bakunts, Andrea Orsi, Data curation, Formal analysis, Validation, Investigation, Methodology; Federica Lari, Resources, Data curation, Investigation, Visualization, Methodology, Writing—review and editing; Laura Tadè, Alberto Danieli, Investigation; Claudia Rato, Investigation, Validation, Writing—review and editing; Caterina Valetti, Conceptualization, Methodology; Roberto Sitia, Resources, Funding acquisition, Writing—review and editing; Andrea Raimondi, Investigation, Methodology; John C Christianson, Conceptualization, Resources, Data curation, Formal analysis, Supervision, Funding acquisition, Investigation, Methodology, Project administration, Writing—review and editing; Eelco van Anken, Conceptualization, Data curation, Formal analysis, Supervision, Funding acquisition, Investigation, Visualization, Methodology, Writing—original draft, Project administration, Writing—review and editing

### Author ORCIDs

Milena Vitale (iD) https://orcid.org/0000-0001-7007-402X
Anush Bakunts (iD) http://orcid.org/0000-0001-8793-1999
Andrea Orsi (iD) https://orcid.org/0000-0003-2839-1640
Federica Lari (iD) http://orcid.org/0000-0003-2789-7877
Claudia Rato (iD) https://orcid.org/0000-0002-3971-046X
Caterina Valetti (iD) http://orcid.org/0000-0003-2917-1586
Roberto Sitia (iD) http://orcid.org/0000-0001-7086-4152
Andrea Raimondi (iD) http://orcid.org/0000-0002-4563-386X
John C Christianson (iD) https://orcid.org/0000-0002-0474-1207
Eelco van Anken (iD) http://orcid.org/0000-0001-9529-2701

### Decision letter and Author response

Decision letter https://doi.org/10.7554/eLife.41168.023

Author response https://doi.org/10.7554/eLife.41168.024

---

## Additional files

### Supplementary files
• Supplementary file 1. List of cell lines and reagents.
DOI: https://doi.org/10.7554/eLife.41168.019
• Transparent reporting form
DOI: https://doi.org/10.7554/eLife.41168.020

### Data availability
All data generated or analysed during this study are included in the manuscript and supporting files.

---

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
