## [Decision Letter]

Thank you for submitting your article "Inadequate BiP availability defines endoplasmic reticulum stress" for consideration for publication as a Research Advance in *eLife*. Your article has been reviewed by three peer reviewers, and the evaluation has been overseen by Peter Walter as Reviewing Editor and Randy Schekman as the Senior Editor. The reviewers have opted to remain anonymous.

The reviewers have discussed the reviews with one another and the Reviewing Editor has drafted this decision to help you prepare a revised submission.

Summary:

Vitale et al. follow up on their earlier paper (Bakunts et al., *eLife* 2017) in which they described a new model to study ER stress. The model entails HeLa cells transfected with inducible IgM heavy-chain μs in presence or absence of λ light chain. Without light-chain, μs expression causes what the authors refer to as "proteostatically driven ER stress", by contrast to less direct ER stress induced by drugs such as thapsigargin or tunicamycin, which is then resolved via the unfolded protein response (UPR). In the present study, they use this model to examine the relative importance of the major ER chaperone, BiP (known to be critical for IgM secretion by plasma cells), and the three UPR branches, IRE1, PERK and ATF6, in the successful resolution of ER stress.

Essential revisions:

1) The authors need to show that BiP's sequestration to μs is the major cause of cytotoxicity as claimed by demonstrating that the sole over-expression of BiP reverts the phenotype. This is important as other ATF6α target genes such as PDI and GRP94 might also impact the cellular fitness.

2) The knockdown levels or knockout phenotype of the transducers needs to analyzed by blotting and be included in the figures.

3) The physiological or pathophysiological relevance of the findings, e.g., for IgM-producing plasma cells, for other specialized secretory cell types, or more broadly to any cell that folds proteins in its ER, remains unknown. As such, some of the conclusions in the title and text are overstated. The effect of μs expression on Ig-producing plasma cells (such as a multiple myeloma cell line) should be examined.

4) The claims of the paper need to be tuned down to match more closely what was actually shown. For example, this paper does not distinguish between the two main models of UPR activation as implied in the Introduction. The chosen client binds to both Ire1 and BiP. Furthermore, the discussion regarding mouse KO phenotypes imply that ATF6 is more generally important in diverse cell types than IRE1 or PERK. While this may be true, the model used here to underscore ATF6 may be more specifically relevant to IgM production by plasma cells than to other types of cells. In the same vein, the authors should qualify more clearly that their conclusions are limited so far to the context of their model system, and may or may not be more generally applicable depending on further studies in other models and contexts.

---

## [Author Response]

Essential revisions:1) The authors need to show that BiP's sequestration to μs is the major cause of cytotoxicity as claimed by demonstrating that the sole over-expression of BiP reverts the phenotype. This is important as other ATF6α target genes such as PDI and GRP94 might also impact the cellular fitness.

We agreed with the reviewers that it would be of interest to over-express BiP. To that end we placed BiP under control of doxycycline (Dox) in the HeLa-µ_s_ cells. Since µ_s_ is the most highly expressed transcript in these cells when they are induced with mifepristone (Mif), over-expression of another gene (i.e. BiP) at the same time leads to attenuation of µ_s_ expression, as there will be competition for the transcription and/or translation machinery, which would confound the interpretation of results. Therefore, we decided to pre-emptively upregulate BiP through Dox before driving µ_s_ expression with Mif. As such, we found that BiP levels could be increased substantially (we estimated it to reach levels of at least 1 order of magnitude higher). Upon subsequent µ_s_ expression (while removing Dox), we witnessed that µ_s_ induction was similar with or without pre-boosted BiP levels, while BiP levels remained highly elevated in the cells that were pre-boosted to express BiP. UPR activation, however, was delayed, as judged by kinetics of XBP1 mRNA splicing, and upregulation of the UPR target GRP94. These results further support that UPR activation primarily reflects how much client protein (µ_s_) builds up in the ER in relation to available BiP. We added a figure displaying these findings and discuss them in the text.

We argue that under conditions of ERAD inhibition, BiP cannot keep pace with µ_s_ levels, which correlates with cytotoxicity. The reviewers argue that if a shortage of BiP indeed leads to cell death that boosting BiP levels should offer protection against cytotoxicity. That idea is attractive, but in practice we obtained preliminary findings that cells suffer in the longer run too when we pre-emptively increase BiP levels and subsequently drive µ_s_ expression (not shown). As is clear from the results described above, cells with pre-emptively boosted BiP levels signal through the UPR with a delay, and thus UPR targets are upregulated with a delay (as shown for GRP94). Apparently, the overall choreography of the response is perturbed in a way that interferes with homeostatic readjustment, but not necessarily similar to the conditions that ensue from BiP running short. We feel that it would not be meaningful to include these findings in the manuscript, as it would lead to convoluted speculations about what the cause(s) of cytotoxicity would be under conditions of a rather artificial form of ER homeostatic failure (i.e. when levels of BiP are preemptively boosted prior to µ_s_ expression). That said, we provide various lines of evidence throughout the manuscript that implicate relative BiP scarcity with cytotoxicity, which is why we feel inclined to discuss that correlation in more detail.

2) The knockdown levels or knockout phenotype of the transducers needs to analyzed by blotting and be included in the figures.

The ablation efficiency of the UPR sensors has been shown as a supplementary figure to Bakunts et al. already, which is cited for this purpose, as is custom. Moreover, from the current figures it is apparent that when IRE1α is ablated ipso facto no splicing of XBP1 mRNA occurs, no CHOP upregulation ensues when PERK is ablated, and that ATF6α itself and upregulation of its downstream targets GRP94 and PDI are undetectable upon ATF6α silencing. We therefore consider that the knockdown efficiency of the three UPR transducers can be reliably inferred from these functional readouts.

3) The physiological or pathophysiological relevance of the findings, e.g., for IgM-producing plasma cells, for other specialized secretory cell types, or more broadly to any cell that folds proteins in its ER, remains unknown. As such, some of the conclusions in the title and text are overstated. The effect of μs expression on Ig-producing plasma cells (such as a multiple myeloma cell line) should be examined.

We and others have studied the effect of IgM production in plasma cells before. The main conclusion from those endeavors pertaining to ER homeostasis (e.g. van Anken et al., 2003) is that ER expansion already precedes IgM production, as it is (at least initially) developmentally driven (i.e. driven by B cell activation). Ectopically inducible expression of µ_s_ in otherwise quiescent B lymphocytes analogous to that in the HeLa-µ_s_ model would be interesting, but B lymphocytes are poorly amenable to genetic engineering and, thus far, our attempts have been unsuccessful. However, it is tempting to speculate that also in B lymphocytes such ectopically induced µ_s_ expression readily would lead to ER stress along similar lines as in the HeLa-µ_s_ model, considering that there is this pre-emptive ER expansion prior to bulk IgM production, apparently to avoid such ER stress.

Most multiple myelomas and/or lymphomas already express Ig subunits in bulk and, consequently, these cells have an ER that is already adapted to the bulk Ig load. For similar reasons as for quiescent B lymphocytes, myelomas and lymphomas are unruly when it comes to their genetic manipulation. For that reason, we exploited some historical mouse myeloma cell lines that were already available in our freezer, namely NS0 (which is famous for being the cell line from which for the first time hybridomas were created by Cesar Milstein and colleagues), and Nµ1. NS0 had been specifically selected to not express any Ig subunits but J-chains (with the hybridoma idea in mind), while Nµ1 was derived from NS0 already in the 1980s having the µ_s_ cassette genomically integrated. The inducible expression system (with sodium butyrate) that was available at the time is less sophisticated than what is currently the standard. Nevertheless, we assessed XBP1 mRNA splicing in both cell lines but there was no splicing in either cell line even upon sodium butyrate treatment. Apparently, these cells (likely due to their being derived from plasma cells) have such a robust ER that further induction of µ_s_ expression posed no detectable ER stress. Accordingly, with a standard tunicamycin dose (5 µg/ml), no splicing was detected in NS0, even after a 4 hr treatment. Yet, the Nµ1 that were induced to expressed µ_s_ showed some splicing at that time point, indicating that tunicamycin and µ_s_ expression synergized to cause ER stress.

Along the same lines, we found that an endometrial cell line (T-HESC), which is another model for professional secretory cells, is highly refractory to tunicamycin treatment (based on the lack of XBP1 splicing – not shown), again likely because they have such an expanded ER already. Moreover, while secretion of various cargo proteins (such as collagens and prolactin) increases when these cells are triggered to decidualize, there is no XBP1 mRNA splicing detectable nor even an increase in BiP transcripts. Thus, cells that have undergone a developmental program to become professional secretors are much more ER stress resistant than non-professional secretory cells (such as HeLa). We feel that these insights have too many ramifications to be included as a control in the current manuscript, and, instead, merit a separate manuscript dedicated exclusively to how ER stress in professional secretory cells may differ from that in non-professional secretors.

4) The claims of the paper need to be tuned down to match more closely what was actually shown. For example, this paper does not distinguish between the two main models of UPR activation as implied in the Introduction. The chosen client binds to both Ire1 and BiP.

We do not distinguish between the two models; our data support both models of UPR activation, which anyway are not mutually exclusive, but instead complementary. We witness through several experimental setups in both Bakunts et al. and the current manuscript that UPR activation is not correlated to the amount of accumulating protein in the ER, nor to the (depletion) levels of BiP per se, as would be the case if either model on its own would explain UPR activation. As we argued already in Bakunts et al., it is the ratio of client/BiP which is the best predictor of UPR activation amplitude, which best fits with a unified model, since the ratio of client bound to UPR sensors versus BiP bound to UPR sensors more robustly reflects the client/BiP ratio than either complex on its own would do. That aside, considering that the two models are complementary, the discussion on which of the two models is correct – in our opinion – is less central, which is why we put this reasoning forward again also in the Introduction of the current manuscript.

Furthermore, the discussion regarding mouse KO phenotypes imply that ATF6 is more generally important in diverse cell types than IRE1 or PERK. While this may be true, the model used here to underscore ATF6 may be more specifically relevant to IgM production by plasma cells than to other types of cells. In the same vein, the authors should qualify more clearly that their conclusions are limited so far to the context of their model system, and may or may not be more generally applicable depending on further studies in other models and contexts.

Upon careful rereading of the particular section in the Discussion, we adapted the text to be more specific. We disagree with the reviewers that the HeLa-µ_s_ model is a model for plasma cells (since neither J nor light chains are expressed). As we envisioned it, the HeLa-µ_s_ model recapitulates how a non-professional secretory cell homeostatically readjusts to the accumulation of a misfolded or orphan client protein in the ER (for instance due to a genetic disease or due to a somatic mutation), which then sequesters the chaperone machinery, in particular BiP (as a multitude of such ill-fated ER clients would do). Certainly, as discussed in Bakunts et al., the HeLa-µ_s_ model better recapitulates such (patho-)physiological events than drug-elicited ER stress does.

We do agree with the reviewers that, depending on the client protein, different type of responses may ensue, which is why we included in the manuscript the non-BiP sequestering µ∆C_H_1, which in fact fails to induce a robust UPR. Moreover, cell types vary in their levels of ER chaperones and UPR transducers, as is also evident from our response above to point 3, but we feel that we would merely state the obvious to suggest that such variability would lead to variability in UPR signaling once these cells are challenged by stressful conditions in the ER.